# Research Trends of Studies on Psychosocial and Health-Related Behaviours of Foreign Domestic Workers in Asia Pacific: A Bibliometric Analysis

**DOI:** 10.3390/healthcare12060690

**Published:** 2024-03-19

**Authors:** Genevieve Ataa Fordjour, Cecilia Lai-Wan Chan

**Affiliations:** 1Department of Psychology, The Education University of Hong Kong, Hong Kong; 2The Jockey Club End-of-Life Community Care Project, The University of Hong Kong, Hong Kong; cecichan@hku.hk; 3Department of Social Work & Social Administration, The University of Hong Kong, Hong Kong; 4Centre on Behavioural Health, The University of Hong Kong, Hong Kong

**Keywords:** psychosocial, health-related behaviours, foreign domestic workers, Asia Pacific, bibliometric analysis, research trends

## Abstract

Foreign domestic workers (FDWs) face challenges that impact their psychosocial well-being and health behaviours. This study utilized bibliometric analyses to examine research trends on the psychosocial and health-related behaviours of FDWs in the Asia Pacific region. The bibliometric analysis comprised citation analysis and co-occurrence analysis. A systematic literature search in academic databases, including Scopus, identified 73 relevant articles published from 1996 to 2023. The growth trend revealed a steady increase in the number of publications on FDWs’ psychosocial and health-related behaviours in Asia over the years, with significant growth from 2018 to 2023, indicating an increasing interest in this research area. The citation analysis identified influential studies, active authors, and sources with high publication numbers in this research area. The analysis also examined the geographical distribution of studies, identifying the countries and organizations in Asia that contributed significantly to FDW research. The co-occurrence analysis of keywords identified key themes and concepts in the literature. The most active keywords identified include “COVID-19”, “Depression”, “Foreign Domestic Workers”, “Mental Health”, and “Quality of Life”. In conclusion, this study provides a comprehensive understanding of the current trends and state of knowledge on the psychosocial and health-related behaviours of FDWs in the Asia Pacific region.

## 1. Introduction

Foreign domestic workers (FDWs) play a vital role in many countries, providing valuable assistance to households by performing various tasks such as cooking, cleaning, childcare, and eldercare. The prevalence of FDWs in Asia Pacific countries is a notable and observable phenomenon. These workers, predominantly from Southeast Asian countries including Indonesia, Philippines, Myanmar, Vietnam, Thailand, Cambodia, and Laos, are in high demand for the provision of domestic assistance and caregiving services in countries like Singapore, Hong Kong SAR, Malaysia, Taiwan, and various others across the region [1,2,3,4]. The increasing demand for FDWs in Asian countries can be attributed to several key factors, including the prevalence of dual-income structures of families [5], ageing populations [2], and the need for additional support in managing household chores [6] as well as caring for children [7], elderly [4], family members [8], or people with disabilities [9].

The duties of FDWs contribute to the smooth functioning of households and enable family members to focus on their work and other commitments [1,8]. Thus, FDWs work helps working parents and family members maintain a work–life balance. FDWs not only ensure the well-being and safety of children in the absence of their parents but also form strong bonds with them [7]. In terms of eldercare, FDWs provide essential support to ageing family members with limited mobility, health conditions, or specific needs [1,4] by assisting with activities such as bathing, dressing, and administering medication. Additionally, FDWs provide emotional support and companionship, enhancing the overall quality of life for the elderly [10].

While FDWs significantly contribute to the well-being of households, they face a myriad of challenges themselves. Leaving their home countries in pursuit of better employment opportunities, many FDWs often leave behind their own families. This separation can lead to feelings of homesickness, isolation, and loneliness as they navigate working in a foreign country [11,12]. In addition, FDWs may encounter challenges, including long working hours [2], low wages [13], language barriers [14], cultural disparities, discrimination [14,15], limited privacy [12], unequal treatment and limited access to healthcare services [3]. These difficulties can have detrimental effects on the psychosocial well-being and health-related behaviours of FDWs, leading to increased stress levels, mental health problems, poor nutrition, sedentary lifestyles, and inadequate healthcare utilization.

It is crucial to understand and address these issues to ensure the overall welfare of FDWs in the Asia Pacific region. Research on the psychosocial and health-related behaviours of Foreign Domestic Workers (FDWs) in the Asia Pacific region has gained increasing attention in recent years [16,17,18,19]. Several studies have examined the unique challenges faced by FDWs in terms of their mental health [17], social well-being [18,19], health-related behaviours [16] and overall health status [20]. Conducting research in this field provides invaluable insights into the experiences and needs of FDWs, which can inform the development of supportive policies, interventions, and programs aimed at promoting their well-being and safeguarding their rights.

Given the growing number of studies on FDWs over the years, it is important to assess the impact and influence of these publications within the community. However, to date, no bibliometric analysis has been conducted to review the literature on the psychosocial and health-related behaviours of FDWs. Analyzing research trends related to the psychosocial and health-related behaviours of FDWs is essential in identifying knowledge gaps and areas that require attention. In this study, we addressed this research gap by conducting a bibliometric analysis, which includes citation analysis and co-occurrence analysis, to comprehensively explore research trends in this area. This study aims to contribute to the current understanding of FDWs’ experiences in the Asia Pacific region, thereby establishing a basis for further advancements in research and support for this workforce. The findings from this study can guide future research endeavours and the necessary intervention strategies to support this workforce.

## 2. Materials and Methods

### 2.1. Data Source

The literature search process was conducted using the Scopus database, which is known for its comprehensive coverage of academic literature. Additionally, a snowballing technique was employed to find additional relevant articles. The snowballing technique involved examining the reference lists and citations of already identified articles to identify any missed sources.

### 2.2. Search Query

The search query employed a combination of relevant keywords, ensuring the inclusion of studies on the psychosocial and health-related behaviours of FDWS in Asia Pacific region countries.

The search query used in the Scopus database was as follows: “TITLE-ABS-KEY (“Psychosocial health” OR “Mental health” OR “Psychological health” OR “Occupational health” OR “Health” OR “Wellbeing” OR “Well-being”) OR TITLE-ABS-KEY (“Social support” OR “Social relationship” OR “Social capital” OR “Coping strategies” OR “Social”) OR TITLE-ABS-KEY (“Healthcare” OR “Health seeking behavio*” OR “Health related”) AND TITLE-ABS-KEY (“Foreign domestic workers” OR “Domestic helpers” OR “Foreign caregivers” OR “Foreign maids” OR “Foreign domestic helpers” OR “ethnic minority” OR “ethnic minorities” OR “caregivers” OR “domestic worker*” OR “migrant workers”) AND TITLE-ABS-KEY (“Asia*” OR “China” OR “Japan” OR “South Korea” OR “India” OR “Indonesia” OR “Philippines” OR “Pakistan” OR “Bangladesh” OR “Vietnam” OR “Turkey” OR “Iran” OR “Thailand” OR “Myanmar” OR “Iraq” OR “Afghanistan” OR “Saudi Arabia” OR “Uzbekistan” OR “Malaysia” OR “Yemen” OR “Nepal” OR “North Korea” OR “Singapore” OR “Sri Lanka” OR “Syria” OR “United Arab Emirates” OR “Cambodia” OR “Lebanon” OR “Kazakhstan” OR “Israel” OR “Jordan” OR “Qatar” OR “Turkmenistan” OR “Azerbaijan” OR “State of Palestine” OR “Armenia” OR “Tajikistan” OR “Mongolia” “Kuwait” OR “Georgia” OR “Laos” OR “Kyrgyzstan” OR “Maldives” OR “Bahrain” OR “Timor-Leste” OR “Oman” OR “Cyprus” OR “Bhutan” OR “Brunei” OR “Hong Kong” OR “Taiwan” OR “Macao”).

### 2.3. Data Selection

The inclusion criteria for article selection were focused on studies that specifically examined the psychosocial and/or health-related behaviours of FDWs in Asia. Studies conducted in non-Asian countries were excluded. This ensured that the analysis was focused on the relevant context of the research. The study was limited to articles only, excluding other document types such as conference papers, review papers, and book chapters. No restrictions were placed on the time span or language.

Figure 1 presents the systematic workflow of this study, which incorporates the various stages involved, from the initial literature search procedure to the subsequent stages of bibliometric analysis and qualitative discussion.

As shown in Figure 1, initially, 654 articles were identified via the Scopus database search. Via snowballing, 24 additional articles were discovered, which were then included in the Scopus database. After removing duplicates, the total number of articles was reduced to 675 articles. These articles were screened by their titles and abstracts, resulting in the exclusion of 528 articles that were deemed irrelevant to the research topic. A total of 63 articles were excluded as they were not conducted in Asian countries, which was a specific inclusion criterion of this study.

Finally, a total of 73 articles were selected for inclusion in the analysis. Table 1 presents an overview of these 73 included studies, detailing the authors’ names and year of publication. 

These articles were deemed relevant to the research topic and aligned with the objectives of the study. The bibliographical information, citation information, abstract, keywords, funding details, and all other information were exported from the Scopus database in CSV file format.

### 2.4. Publication Growth Trend Analysis

This study sought to explore the research trend of studies on FDW. Research trend analysis plays a crucial role in understanding the current state of research in any field. Examining the number of publications published on a yearly basis allows researchers to estimate the publication growth trend and make predictions about future trends. In this study, a descriptive analysis was conducted to determine the yearly number of publications on FDW studies and the cumulative publications on a year-to-year basis. Via the publication growth trend analysis, insights into the research trend in the field of FDW studies can be obtained, providing a foundation for identifying knowledge gaps and directing future research.

### 2.5. Bibliometric Analysis

Bibliometric analysis, which is a statistical evaluation of published documents, serves as an effective way to measure the influence of publications, authors, sources, countries, or institutions within the scientific community [86]. In this study, bibliometric analysis was conducted using the VOSviewer version 1.6.20 software developed by van Eck and Waltman [87]. This software provides a virtual representation of bibliometric data via comprehensive and easily interpretable maps. The maps generated from the bibliometric analysis provide visual representations of relationships, facilitating exploration of not only author activity but also interconnectivity with other authors, sources, countries, and organizations [88,89]. Notably, the distances between nodes within the bibliometric map reflect the closeness between the relationships [87].

Employing the VOSviewer software, we generated and analyzed various bibliometric maps based on citation analysis and co-occurrence analysis of keywords from the 73 included studies. Citation analysis enabled the identification of relational networks among the 73 included studies and the assessment of their impact within the scientific community based on the number of times they were cited by other authors. The impact of documents, authors, countries, organizations, and sources was measured, while the citation network revealed the citation relationships between authors and documents. Meanwhile, the co-occurrence analysis of keywords was conducted to identify the most active keywords explored within the field of FDWs.

Bibliometric analysis was employed in this study to provide a statistical evaluation and visualization of the research trend in the field of studies on FDWs in the Asia Pacific region, thereby offering a unique perspective that complements traditional review methods such as systematic review or meta-analysis [86,89]. Overall, these bibliometric analyses provide valuable insights into the relationships, impact, and keyword trends within the field of FDWs, ultimately contributing to a deeper understanding and advancement of research in this area.

### 2.6. Narrative Review

The analysis of top-cited studies involved a narrative review that delved into the research themes, the countries where the studies were conducted, the study designs employed, and the key findings. This comprehensive evaluation allowed for an in-depth understanding of the psychosocial and health-related behaviours of FDWs in Asia. Thus, by examining these aspects across the selected articles, valuable insights were gained into the diverse perspectives and trends shaping research in this field.

## 3. Results

Only studies conducted in Asia were included in the research selection process to ensure that the analysis focused on the context of psychosocial and health-related behaviours of FDWs in the Asia Pacific region. By narrowing the scope to Asian countries, the study aimed to capture the unique context and challenges faced by FDWs in this specific geographical area. This approach thus allowed for a more targeted exploration of the research literature that directly pertains to the experiences and needs of FDWs in Asia.

### 3.1. Growth Trend of Journal Publications on FDW

The number of publications and cumulative publications on a yearly basis for the 73 included studies on psychosocial and health-related behaviours of foreign domestic workers (FDWs) in Asia were assessed to provide a clear estimate of the research growth trend in the subject area, which has been depicted in Figure 2. The plotted line graph of the cumulative publications clearly depicts the growth trend of research publications in this field.

The study discovered that article publications on the psychological well-being and health-related behaviours of FDWs can be traced back to 1996, but significant interest from researchers was observed starting in 2015. Since 2018, at least four articles have been published annually on FDWs. The year 2022 witnessed the highest number of publications, totalling 17 articles. As of November 2023, this year, 12 articles have been published in this research area, bringing the total number of articles in this study to 73.

Additionally, the cumulative publication graph clearly showcases the exponential growth in article publications on the psychological well-being and health-related behaviours of FDWs over the years. This research trend effectively indicates the increasing attention that researchers worldwide have dedicated to this topic from 1996 to 2023. These findings not only highlight the growing interest in and importance of research on FDW but also emphasize the need for further investigation and support in this field. The significant increase in publications also implies a growing recognition of the unique experiences and needs of FDWs, which can contribute to the development of supportive policies, interventions, and programs aimed at promoting their well-being and safeguarding their rights.

### 3.2. Results from Bibliometric Analysis

#### 3.2.1. Citation Analysis of the Documents

The citation analysis of the documents provides information on the quality of the published document. A publication with a higher citation metric indicates that the quality of the document is high and has been cited by many researchers [90]. The citation relationship network between the 73 included studies on FDW was mapped as presented in Figure 3. 

The map generated from the bibliometric analysis revealed that out of the 73 included studies on FDW, only 8 documents had citation connections [12,17,23,43,48,49,69,74], while the remaining 65 documents did not have any citation connections. These 65 documents, including Ho et al. [66] and Wong et al. [38], were excluded from the bibliometric map in Figure 3. As shown in Figure 3, the documents formed 4 clusters, with the largest group consisting of documents including Lu et al. [49] and Su et al. [74]. 

This study further sought to identify the highly cited documents from the 73 included literature review documents on FDW, with at least 10 citations. A total of 20 documents met this criterion. Table 2 presents the list of these highly cited documents.

The top-cited article was Holroyd et al. [23], which reported the health-related behaviours, health locus of control and social support of Filipino domestic workers in Hong Kong. This article, published in 2001, has received 51 citations and has an average citation rate of 1.00 per the year 2001. The study by Holroyd et al. [23] had the largest citation links with four studies, namely Mok and Ho [12], Hall et al. [17], Bernadas and Jiang [43], and Cheung et al. [69].

The second most cited article was by Hall et al. [17], titled “The effect of discrimination on depression and anxiety symptoms and the buffering role of social capital among female domestic workers in Macao, China”. Published in 2019, this article has received 48 citations and has an average citation rate of 0.44 per the year 2019. Other notable articles include “Knowledge, attitudes and practices towards COVID-19 amongst ethnic minorities in Hong Kong” by Wong et al. [16], “Abuse and depression among Filipino foreign domestic helpers. A cross-sectional survey in Hong Kong by Cheung et al. [69], and “Dancing to different tunes: Performance and activism among migrant domestic workers in Hong Kong” by Lai [32]. These articles have received 36, 35, and 33 citations, respectively, and contribute to our understanding of the experiences and challenges faced by FDW in Asian countries.

Overall, these highly cited articles highlight the importance and relevance of research on domestic workers and shed light on various aspects of their lives, including health, well-being, social support, and discrimination. The number of citations reflects the significant impact these studies have had on the field and the recognition they have received from the research community.

#### 3.2.2. Authors and Co-Authors Relationship

The analysis of the authors’ and co-authors’ relationship helps to identify the major research scholars who are working in a particular field [89]. Using citation analysis, the authors’ and co-authors’ relationship network was easily mapped and has been presented in Figure 4.

The bibliometric analysis of the 73 review documents revealed 215 authors and co-authors, of which 36 were interconnected. Figure 4 divulged that there were four author groups, each with at least six authors who have published studies in the field of FDW. The research groups of authors, including Su H.-C. and Chang J.-S., were the prominent groups that have published review studies related to FDW.

Though there might not be a direct relationship between the researchers, they are invariantly connected to each other as they share their research expertise in similar fields of interest. The researchers are also connected to each other via citation networks, the content of their published documents, and publication outlets, among other factors. Interestingly, as shown in Figure 4, the research groups were closely connected and collaborated with other research groups. It can be concluded from the discussion that there is extensive research collaboration on FDW’s research among the different research groups. This allows for knowledge flow among the different research groups and helps improve the quality of the research on FDW’s psychosocial and/or health-related behaviours in Asia.

The number of publications on FDW by the authors and the citation metrics received by the authors were used to identify the most active researchers in the field of research. Table 3 presents the top authors with at least two publications on FDW, totalling 17 authors.

Among all the authors and co-authors, So W. K.W., Wong C. L. and Arat G. have the highest number of publications, with four each, on FDWs’ psychosocial and/or health-related behaviours in Asia. Both So W. K.W. and Wong C. L. also have the highest citation counts of 62 each. However, with two documents each, Chan C.W.H. and Chow K.M. have the second-highest citation counts, with 61 each. Although Arat G. has also published the highest publication of 4 documents, the accumulated citation count is relatively lower, with only 19 counts.

Further analysis, using the average citations per document, provides more information on the impact of the researchers. Leung A.W.Y., with two published articles on FDW, has the highest average citations per document of 21.00. This was followed by Chan C.W.H. and Chow K.M., both with the second-highest average citations per document of 20.33. This indicates that the documents of Leung A.W.Y., Chan C.W.H. and Chow K.M. have made a greater impact compared to the other researchers in Asia. Interestingly, So W.K.W. and Wong C.L., despite having the highest number of publications on FDWs’ psychosocial and/or health-related behaviours literature, are both ranked fifth in terms of average citations per document, with 15.50. The total link strength also reveals that So W. K.W. and Wong C. L. have the highest research collaboration, making them prolific researchers in this field of research. It is generally believed that good-quality publications receive higher numbers of citations. The publications with relatively low citation numbers are likely due to the timeline of the publications. Besides the time factor, other factors, such as research collaboration and the strength of the network, can also increase researchers’ influence in the scientific community. Therefore, extensive research collaboration is necessary to enhance the knowledge flow and improve the quality of work on FDWs’ psychosocial and/or health-related behaviours in Asia.

#### 3.2.3. Source Distribution and Citation Relationship

The analysis of the distribution of publications among the various sources and their citation relationships provides an indication of the journals or outlets in which the researchers usually publish their studies on FDW. The bibliometric analysis conducted revealed 56 sources from the 73 documents explored in this study. Figure 5 presents the map generated from the bibliometric analysis, depicting the citation network of the publication sources.

The bibliometric map revealed that out of the 56 sources, there were 17 sources that had citation connections with other sources. The remaining 39 sources, which had no connections, were excluded from the map shown in Figure 5. This study further sought to identify the top sources that have published articles related to FDW. Table 4 enlists the top journals, along with their respective impact factors.

It can be observed from Table 4 that the International Journal of Environmental Research and Public Health (IJERPH) was the preferred choice of researchers in Asia for publishing their studies on FDW. From this source, 13 articles on FDW have been published to date, and these documents have accumulated 93 citations so far. The second most preferred choice was the Asian and Pacific Migration Journal, with three published articles and 20 citations. In terms of average citations per document, the IJERPH had the highest number, 7.15, out of the 13 published articles on FDW, indicating the impactful nature of the articles published in IJERPH. This was followed by the Journal of Applied Gerontology, with an average of 7.00 citations per document.

Based on this data, we can conclude that the top sources of high-quality research on FDWs in Asia were IJERPH and the Journal of Applied Gerontology. This can be substantiated by the fact that the impact factors of these journals were above 3.00. It is also noteworthy that IJERPH and Asian Education and Development Studies had the highest total link strength of 2 each. This suggests that these sources were highly cited by the documents published in the other sources, which can be confirmed by referring to the source-citation relationship map in Figure 5.

#### 3.2.4. Distribution of Publications on a Country Level

The 73 articles on FDW were published from 19 different countries. The total number of publications by adding the contributions from each of the 19 countries is 91, which exceeds the 73 number of documents under review. This indicates that there has been collaborative work between different countries in conducting FDW studies.

The bibliometric analysis depicted in Figure 6 revealed that out of the 19 countries in Asia that have published at least one document on FDW, only eight countries have had research collaborations. The countries without any research collaborations were excluded from the mapping. Furthermore, the analysis shows that Hong Kong SAR has published documents on FDW in collaboration with six other countries/territories in Asia, including mainland China, Macau, Japan, Taiwan and Vietnam. The statistics from the bibliometric analysis of the 19 countries are provided in Table 5.

The highest number of article publications on FDW in Asia (37 articles, which is 41.11% of the total documents) was from Hong Kong SAR. Taiwan shared the second position, publishing 16 articles (17.78% of the total documents), followed by nine articles (10.00%) from mainland China. Figure 7 presents a virtual description of the top 6 countries that published at least three articles related to FDWs’ psychosocial and health-related behaviours in Asia.

In general, there were five top countries that published at least three articles related to FDWs’ psychosocial and health-related behaviours in Asia. It is evident also from Table 5 that out of 19 countries, five countries rank among the top 20 countries in the world in terms of nominal GDP. This indicates that economically developed countries have recognized the importance of FDWs’ well-being. Additionally, according to the bibliometric data, Hong Kong SAR received the highest number of 387 citations from the 37 articles on FDWs published in the country. Interestingly, the average citation of Hong Kong SAR was in the second position (10.46), with Macau taking the first position according to the highest average citation of 26.00 obtained from a total of two documents published on FDWs.

The Total Link Strength (TLS) provides an estimate of the collaborative research between the countries, as shown in Table 5. Although Hong Kong SAR had the highest number of research collaborations with 14 countries, the analysis of TLS suggested that Taiwan was by far the most prominent country in terms of collaborative research on FDWs’ psychosocial and health-related behaviours in Asia, with a TLS of 7. The bibliometric analysis revealed that the researchers from Taiwan had published documents on FDW in collaboration with seven other countries/territories in Asia, such as India, Thailand, Indonesia, Vietnam, and Hong Kong SAR. With a TLS score of 6, the Hong Kong SAR was in the second position for collaborative research in this field of study. From the analysis of the TLS score and countries’ collaboration network map, it is evident that Taiwan, Hong Kong SAR and Indonesia have robust research collaboration on FDWs studies.

#### 3.2.5. Organizations with Highest Publication on FDW

The bibliometric analysis revealed that 102 organizations were affiliated with the 73 included studies on FDW. Figure 8 presents the citation relationship network of the affiliated organizations.

The mapping in Figure 8 reveals that out of the 104 organizations, only 18 organizations (highlighted with bright colours) have citation connections with each other for this field of study, while 86 organizations have no connections with other organisations. In gist, it can be concluded that the articles from the organisations in Hong Kong have had the most significant impact on the studies regarding FDWs. The top organizations that have published at least three studies on FDWs are identified and presented in Table 6.

The bibliometric analysis shown in Table 6 reveals that only six organizations have published at least three articles on FDWs’ psychosocial and/or health-related behaviours over the years. It was found that the Chinese University of Hong Kong has published the highest number of 12 articles on FDWs, with a total of 249 accumulated citations. This results in an average of 20.75 citations per document. The analysis also revealed that the University of Hong Kong, with the second highest number of 11 publications, has accumulated 102 citation counts. Although the University of Macau was identified with only two articles on FDWs, these documents have accumulated a total of 52 number of citations, with the highest average citation of 26.00 per document. The total link strength also revealed that the Chinese University of Hong Kong has been cited 14 times by these top twelve organisations. From the analysis, it can be concluded that the Universities in Hong Kong, Taiwan, Macau, Thailand and Indonesia play a dominant role in this area of research.

#### 3.2.6. Identifying the Most Active Keywords for FDW Literature

A co-occurrence analysis of all keywords was conducted to identify the most commonly used keywords by researchers. The bibliometric analysis revealed that a total of 763 keywords were used in the 73 included studies, with 273 being author keywords. Figure 9 presents a network visualization map of the keywords with two co-occurrences, comprising a total of 228 keywords.

Additionally, a density visualization map was generated from the author’s keywords with a minimum of 2 occurrences, as depicted in Figure 10.

The analysis of the density visualization map highlighted 38 author keywords with a minimum of two occurrences that were commonly reflected in the 73 studies on FDWs. Furthermore, the science mapping of the most active keywords generated six clusters, with each representing a relevant domain such as quality of life, depressive symptoms, public health and COVID-19. The statistical details of the most active author keywords with three minimum occurrences have been presented in Table 7.

The results indicate that the most repeated keywords were ‘COVID-19’, followed by ‘depression’ and ‘ethnic minority’. The total link strength, which defines the strength of interrelatedness between keywords, also revealed ‘COVID-19’ and ‘depression’ as the most active keywords. Hence, it can be concluded that ‘COVID-19’ and ‘depression’ are the main themes of the studies on FDWs’ psychosocial and health-related behaviours.

### 3.3. Results of Narrative Review

#### 3.3.1. Research Themes

The research themes from these 73 included studies encompass a wide range of topics related to the experiences, challenges, and well-being of FDWs in Asia. These research themes address various aspects of FDWs’ lives and provide valuable insights into their psychosocial and health-related behaviours. One of the prominent themes identified from the included studies is mental health, with several studies focusing on issues such as depression, anxiety, and other depressive symptoms among FDWs [72,75,89]. The studies aim to understand the prevalence, factors, and impacts of mental health issues on FDWs and highlight the need for appropriate support and interventions [28,53].

Religious transformation is another notable theme in the literature, exploring how FDWs’ religious beliefs and practices may change while working in a foreign context. Rohmaniyah and colleagues [58] examined the role of religion in coping mechanisms, personal identity, and social integration among FDWs. Community health services utilization is another important area of research investigating the access and utilization of healthcare services by FDWs. Kwan and Lo [56] identified barriers and facilitators to healthcare access and improve the quality of healthcare services available to FDWs. Language learning is another theme that emerged from the literature, examining the challenges and strategies adopted by FDWs in acquiring the language skills necessary for effective communication and integration in their host countries [15].

Caregiver burden [63] and social support [18] are also widely explored topics, focusing on the challenges faced by FDWs in their caregiving roles and the support networks available to them. These studies shed light on the emotional and physical strain experienced by FDWs and the importance of social support in mitigating their challenges [18,63]. The impact of the COVID-19 pandemic on FDWs is also a significant theme identified in the literature. Wong et al. [16] examined the unique challenges faced by FDWs, such as increased vulnerability, social isolation, and workload during the pandemic. This research helps to inform policies and interventions to address the specific needs of FDWs in times of crisis [16].

Other research themes identified include social space utilization [12], minimum wages [13], cultural intelligence [15], psychological morbidities [28], reproductive health education [82], stigma [84], migration issues [35,39], financial implications of borrowing [78], struggles and activism [41], rehabilitation services [81], intergenerational caregiving, and childcare provision [63], subjectivities [55] and health-related behaviours [20].

These studies contribute to a comprehensive understanding of the complexities and needs of FDWs in Asia. By addressing a wide range of topics, these studies provide valuable insights into the multifaceted lives of FDWs and serve as a foundation for future research and support programs aimed at improving their well-being and rights.

#### 3.3.2. Key Findings from Top-Cited Studies

This study delves into the key findings from the 20 highly cited studies identified using the bibliometric analysis, shedding light on the country of study, study type, study participants, and significant insights regarding the psychosocial and health-related behaviours among FDWs in the Asia Pacific region. Table 8 presents these findings.

#### Findings on Psychosocial among FDWs in Asia

The research conducted by Holroyd and colleagues found that Filipino domestic workers in Hong Kong frequently experience psychosocial distress symptoms, including waking up early, feelings of loneliness, worry, and difficulty falling asleep [23]. The study by Hall and colleagues revealed that Filipino domestic workers in Macao, China, experience a higher burden of mental disorders. Discrimination was identified as a significant factor contributing to this burden. Additionally, their study indicated that social resources, specifically cognitive and social capital, can act as a buffer against the negative impacts of discrimination. Correlation analyses highlighted a significant association between discrimination, depression, and anxiety. Cognitive–social capital was found to have a negative association with depression and anxiety. Moreover, the study found that cognitive–social capital modified the relationship between discrimination, depression, and anxiety. Surprisingly, individuals with moderate to high levels of cognitive–social capital experienced worsening symptoms as discrimination increased, challenging previous assumptions about the protective role of social resources in mental health [17].

The study conducted by Cheung and colleagues in Hong Kong, China, revealed that among Filipino domestic workers, 20.5% reported experiencing physical abuse, and 34.4% reported verbal abuse in the past 12 months. Female employers were identified as the main perpetrators of abuse. Surprisingly, a significant percentage (16.7%) of abuse victims did not report their cases, with only 19.4% choosing to report to formal organizations such as the police. Their study also found that these experiences of physical abuse and verbal abuse were associated with higher levels of depression among the participants. Additionally, non-disclosure of physical abuse experiences and dissatisfaction with living spaces were linked to elevated depression levels [69]. Lai’s study explored the significance of cultural performances in the activism of Indonesian and Filipina domestic workers in Hong Kong, China. These performances were found to play a pivotal role in building community, creating a collective identity, showcasing worker agency, and highlighting diversity and unity among these workers [32]. Leung and colleagues’ study in Hong Kong, China, explored how social support influences caregiver burden among South Asian domestic workers. While friend support was found to have a significant positive direct effect on caregiver burden, the research highlighted the pivotal role of caregiving self-efficacy in alleviating burden. The study emphasized that sources of social support act as vital moderators in the relationship between caregiving self-efficacy and caregiver burden [18].

The study by Mackenzie and Holroyd in Hong Kong, China, revealed that caregiving significantly impacts the emotional and psychological well-being of South Asian domestic workers, often leading to stress and burden in their caregiving role. Additionally, the study found that the effects of caregiving can diminish the overall quality of life for both caregivers and dependent individuals. Family and kinship relationships were identified as pivotal in caregiving situations, with obligations and feelings of care sustaining the caregiving tasks [21]. Gu and Han’s study on South Asian domestic workers in Hong Kong, China, explored family language planning (FLP) as influenced by diverse social factors, evolving language values, and social and ideological contexts. Interviews with mothers highlighted the importance of language planning within the home, reflecting individual experiences across various social spheres. Their study revealed that the changing values of languages over time and space play a significant role in the psychological well-being of FDWs [19]. Ogaya’s study on Filipina domestic workers in Hong Kong, China, revealed that social activities, such as peer counselling, skills development, and participation in religious and cultural events, are instrumental in empowering migrant domestic workers, increasing their social visibility and enhancing their psychosocial health [25].

Fan and Chen’s study on South Asian domestic workers in Taiwan, China, revealed that the care burden has substantial impacts on caregivers’ physical health, psychological well-being, social relationships, and environmental perceptions. Moreover, caregivers’ care burden is intricately connected to both subjective and objective aspects of their quality of life, irrespective of diagnostic categories or the time spent with patients. The study emphasized that the duration of caregiving, particularly the daily care hours, plays a pivotal role in determining care burden, underscoring the significance of current caregiving demands on caregiver well-being [33]. Wong and Zelman’s study on South Asian domestic workers in Hong Kong, China, revealed that behavioural and psychological symptoms of dementia (BPSD), including agitation, delusions, and irritability, were notably linked to caregiver burden and depression. Their research emphasized that expressed emotion, particularly intrusiveness, played a crucial role in mediating the association between BPSD and adverse caregiver outcomes. Additionally, factors such as caregiving hours, inadequate family support, and lack of religious affiliation were identified as contributors to heightened expressed emotion and poorer caregiver outcomes. The study underscored that the detrimental effects of BPSD on caregivers in Hong Kong were influenced by expressed emotion, highlighting the significance of comprehending family dynamics and social support in dementia caregiving [76].

The study by Chui and colleagues identified that Filipino domestic workers in Hong Kong, China, experienced loneliness, strained family relationships, susceptibility to depression, and early signs of cognitive impairment. Their study underscored the importance of early cognitive impairment screening and interventions to address mental health and social support needs, as well as the significance of community and health services in supporting ageing individuals with chronic illnesses, some of whom faced challenges in returning home independently [83]. The study by Shim and Lee investigated the correlation between risk-taking tendencies and attitudes among Southeast and East Asian domestic workers in China, Japan, Korea, and Taiwan. Their research revealed that various psychosocial factors, such as age, socio-demographics, and the rapid ageing trends in these Asian countries, play a significant role in shaping how FDWs perceive reports, political propaganda, and rumours circulating about migration. These influences further impact individual risk preferences and attitudes towards migration, potentially affecting decisions related to health and behaviour [52]. The study by Lee uncovered that South Asian domestic workers in Taiwan, China emphasized the significance of taking into account psychosocial factors, like role strain and role reward when providing care for individuals with cognitive impairments, as these factors can impact caregiver outcomes and the quality of care given [31].

#### Findings on Health-Related Behaviours among FDWs in Asia

The study conducted by Holroyd and colleagues found that Filipino domestic workers in Hong Kong generally exhibited good health-related behaviours, including a healthy diet and low alcohol, nicotine, and coffee consumption. However, their Pap smear rates were low, as were scores on other preventive health practices. Many of their participants felt their health behaviours depended on chance or influential individuals [23]. The research conducted by Wong and colleagues highlighted that norms and self-efficacy are influential factors in shaping health-related intentions and behaviours among South Asian domestic workers in Hong Kong, China. Additionally, social norms and beliefs were found to impact individuals’ attitudes to health practices, ultimately influencing behaviours [16]. In the study conducted by So and colleagues in Hong Kong, China, the Pap smear test uptake among South Asian domestic workers revealed significantly lower rates compared to the general population, indicating disparities in preventive health behaviours. Various factors were found to independently influence test uptake, including ethnicity, age, education level, marital status, family history of cancer, smoking habits, use of complementary therapies, health-related beliefs, perceived cancer susceptibility, and recommendations from healthcare professionals. Notably, the belief in the benefits of exercise for health had a stronger influence on test uptake among ethnic minority women than in the general population [20].

Vandan and colleagues’ study on South Asian domestic workers in Hong Kong, China, found that participants acknowledged the significance of health, as evidenced by familiar sayings like “health is wealth” and “walking is good exercise.” However, they admitted to falling short in adopting healthy eating and lifestyle practices. The research identified low self-priority among participants, with self-care and health-seeking behaviours not being given precedence. Furthermore, their study noted limited access to health-related information among participants of diverse educational backgrounds, who either depended on family and friends for information or encountered challenges in accessing health resources due to language barriers [61]. Fan and Chen’s research on South Asian domestic workers in Taiwan, China, unveils a substantial correlation between care burden and caregivers’ quality of life, affecting their physical health and well-being. The study highlights that the time devoted to caregiving is a crucial predictor of care burden, signifying that the intensity of daily care hours significantly impacts caregivers’ overall well-being. The findings indicate that the care burden is more influenced by the immediate caregiving challenges perceived by caregivers rather than solely being a consequence of the long-term progression of the illness [33].

Yeung and colleagues’ study on Filipino domestic workers in Hong Kong, China, uncovered that probable anxiety among FDWs was influenced by factors such as increased workload, insufficient protective equipment, and concerns about job insecurity during the COVID-19 pandemic, underscoring the unique stressors faced by this vulnerable group. The research highlighted that employment-related rights, workload intensity, and fears of COVID-19 transmission significantly impacted the psychosocial well-being of FDWs in Hong Kong during the crisis [73]. Ngan and Chan’s study on domestic workers in South Korea, Taiwan, Hong Kong, SAR-China, and mainland China revealed that the social realities encountered by FDWs impact their health outcomes. The research emphasized the need to address issues related to social exclusion concerning institutional labour protection and citizens’ rights, which could potentially impact health-seeking behaviours and access to healthcare services among these migrants [35].

Bernadas and Jiang’s study on Filipino domestic workers in Hong Kong, China, revealed that health information-seeking behaviours are influenced by factors such as individuals’ experiences with a particular condition, perceived efficacy of health information, the salience of health issues, and the importance assigned to specific health concerns. For example, women may seek screening guidelines for breast cancer when they have concerns about it, highlighting the impact of perceived importance and efficacy on health-related behaviours. The research also emphasized that information-seeking behaviour can be driven by personal experiences with health-threatening conditions, social networks, and beliefs about one’s ability to effectively search for health information [43]. The study by Chui and colleagues found that Filipino domestic workers in Hong Kong, China, emphasized the importance of having access to suitable health and social care services tailored to their physical health needs for the effective management of chronic conditions and the enhancement of their overall well-being [83]. The study conducted by Wang and colleagues revealed that foreign domestic workers (FDWs) in Singapore, Hong Kong, and Taiwan observed that the ageing population in these Asian countries has led to an increase in their healthcare and social care responsibilities. These responsibilities play a crucial role in influencing their health-related behaviours to ensure their overall health and well-being [13].

## 4. Discussion

### 4.1. Summary of Findings on Publication Growth Trend and Bibliometric Analyses

This paper discusses the bibliometric information of the published studies related to FDWs’ psychosocial and health-related behaviours. The 73 included studies were extracted from the Scopus database and snowballing references. Bibliometric analysis using VOSviewer Software was performed on the extracted information from the Scopus database. It was observed that the studies related to FDWs started in 1996. However, it was not until 2018 that the studies on FDWs started receiving much attention, with an exponential rise in publications.

The document of Holroyd and colleagues [23] was the most cited publication, with 51 citations. It was revealed that four research groups exist, and collaboration connections exist with at least twenty other researchers who have worked on FDWs’ psychosocial and health-related behaviours. Hence, there is a huge scope of research collaboration across the globe, which in turn helps to improve the quality of research in this field of study. The highest number of documents were published by So W.K.W., Wong C. L. and Arat G. Whilst Chan C.W.H. and Chow K.M. were the most impactful researchers in terms of average citations per document. In terms of research collaboration, So W. K.W. and Wong C. L. were the most influential researchers.

Most of the articles on FDWs were published by the International Journal of Environmental Research and Public Health, with the most impactful documents and the highest average citations per document. It was found that most of the countries involved in FDW research in Asia were economically developed. Hong Kong SAR was the country/territory that published the highest number of articles on FDW. This was followed by Taiwan, which shared the second position. Interestingly, Macau, with two articles on FDWs’ psychosocial and health-related behaviours, was the most impactful country in terms of the highest average citations per document. The results also indicated that Taiwan had the highest research collaboration with other countries.

It was found that the Chinese University of Hong Kong was the most impactful and influential affiliated organisation that had published the highest number of articles on FDWs’ psychosocial and health-related behaviours. The publications from the University of Hong Kong and the University of Macau also received good attention from other researchers, accumulating a high average of citations per document. The results from the bibliometric analysis revealed that the most active keywords in this research domain were COVID-19 and depression.

### 4.2. Summary of Findings on Narrative Review

The top-cited studies were conducted in various Asian countries, including Hong Kong SAR-China, Macao-China, Singapore, Taiwan-China, South Korea, Mainland China, Japan, and Korea, where psychosocial and health-related behaviours among FDWs were examined. The findings suggest a significant presence of Filipinos and South Asians as FDWs in Asia. The included studies utilized various research methodologies to address psychosocial factors and health-related behaviours among the FDWs, including cross-sectional surveys, interviews, and semi-structured interviews.

The research conducted on FDWs in various Asian countries highlights the prevalence of psychosocial distress, discrimination, abuse, caregiving challenges, and mental health issues within this population. The studies identified explored the impact of factors such as social support [18,76], cultural performances [32], language planning [19], caregiver burden [21], loneliness [74], discrimination [17], self-efficacy [16], verbal abuse [49] on the psychosocial well-being of FDWs. These findings underscore the importance of addressing psychosocial needs, enhancing social support systems, and implementing interventions to improve the mental health and overall quality of life of migrant domestic workers in Asia. The findings from various studies on health-related behaviours among foreign domestic workers (FDWs) in Asia also reveal several key insights. Filipino domestic workers in Hong Kong were found to exhibit good health-related behaviours, such as a healthy diet and low alcohol consumption [23]. However, their Pap smear rates were low, indicating disparities in preventive health behaviours [23]. Factors such as social norms [16], beliefs [20], access to health information [24,25], social exclusion [35], labour protection [35], and healthcare access [23] influenced the health behaviours of FDWs in various Asian countries. Additionally, the importance of tailored health and social care services for FDWs and the impact of ageing populations [13,83] on their healthcare responsibilities were highlighted in these studies. Overall, addressing factors such as access to information, social support, and work-related stress is crucial in promoting the health and well-being of FDWs in Asia.

Further research studies have also identified that FDWs often face obstacles such as extended working hours [2], language barriers [14], cultural disparities, discrimination [14,15], and limited privacy [12]. These challenges can significantly impact the psychosocial well-being and health-related behaviours of FDWs, resulting in elevated stress levels, mental health issues, poor nutrition, sedentary lifestyles, and insufficient utilization of healthcare services.

### 4.3. Research Gaps

The literature reveals a substantial body of research focused on the psychosocial and health-related behaviours of foreign domestic workers (FDWs) in the Asia Pacific region. Previous studies have undertaken scoping reviews, systematic reviews, and meta-analyses examining various aspects of FDWs’ well-being. For instance, a scoping review delved into the health stressors, problems, and coping mechanisms of migrant domestic workers worldwide [93]. Additionally, Ho and colleagues [94] explored peer support and mental health among migrant domestic workers globally via a scoping review. Perski and colleagues [95] conducted a systematic review and meta-analysis of ecological momentary assessment studies on five key health behaviours on a global scale, albeit not specifically focusing on FDWs. Notably, there was a lack of bibliometric reviews in the existing literature. This study aims to address this gap by conducting a comprehensive bibliometric analysis of the psychosocial and health-related behaviours of FDWs in the Asia Pacific.

Via this study, several gaps in the existing literature have been identified, shedding light on areas that warrant further research and scrutiny. Firstly, many of the studies identified are cross-sectional in nature, providing a snapshot of the psychosocial and health-related behaviours of foreign domestic workers. There is a need for longitudinal studies that follow these workers over an extended period to assess changes in their behaviours and experiences over time. Secondly, while some studies touch upon psychosocial health issues, such as depression and anxiety symptoms, there is a potential gap in the literature regarding a comprehensive exploration of psychosocial factors among FDWs. Further research can focus on understanding the psychosocial factors and strategies to support their psychological well-being.

Thirdly, the existing studies may not explicitly explore the influence of cultural factors on FDWs’ psychosocial and health-related behaviours. Investigating cultural influences, such as cultural norms and values, acculturation processes, and cultural competency, can provide a deeper understanding of the unique challenges faced by these workers and their impact on their well-being. Additionally, while there are a few studies discussing policy implications, such as minimum wage regulations and legal protection, further research could delve into the effectiveness and enforcement of existing policies and identify gaps in providing adequate support and protection for foreign domestic workers. Furthermore, the studies identified primarily focus on the experiences and behaviours of FDWs as a general group. However, there is a need for research that considers the intersecting identities and unique experiences of different subgroups within the foreign domestic worker population, such as based on ethnicity, nationality, and/or gender.

Addressing these literature gaps can contribute to a more comprehensive understanding of FDWs’ psychosocial and health-related behaviours in the Asia Pacific region and inform policies and interventions aimed at promoting their well-being.

### 4.4. Practical Implications

The generated bibliometric maps and analyses offer valuable guidance for researchers and academics interested in the field of FDWs. By identifying the most active keywords, impactful documents, authors, and organizations, this information can guide researchers on the current trends, key areas of focus, and potential collaborators within the field. Understanding the citation relationships and impact of studies can help direct future research endeavours and collaborations. Additionally, identifying the most discussed topics and trends related to FDWs’ psychosocial and health-related behaviours can aid in formulating evidence-based policies, interventions, and programs aimed at improving the well-being and rights of FDWs in the Asia Pacific region. Thus, researchers, practitioners, and policymakers can leverage these highly cited articles and impactful studies to build upon existing knowledge, promote collaboration, and advocate for the rights and well-being of FDWs.

## 5. Conclusions

The findings from this study provide valuable insights into the research trends and current state of knowledge regarding the psychosocial and health-related behaviours of foreign domestic workers (FDWs) in the Asia Pacific region. Using the application of bibliometric analysis, the aim of this study was to comprehensively explore the research landscape in this area and identify knowledge gaps that require attention. The scope of the study focused specifically on studies conducted in Asia, ensuring that the analysis remained contextually relevant to the experiences of FDWs in the region. A literature search identified a total of 73 articles that met the inclusion criteria, providing a comprehensive dataset for further examination and discussion.

The analysis revealed a growing number of publications in FDW studies over the years, indicating an increased interest in understanding the experiences and needs of this workforce. This trend suggests recognition of the importance of addressing the psychosocial and health-related aspects of FDWs’ lives. The included studies shed light on pivotal insights concerning the psychosocial and health-related behaviours of FDWs in the Asia Pacific region. Notably, psychosocial factors such as engaging in social activities beyond their employers’ households were identified as crucial determinants of the well-being of FDWs, emphasizing their significance on FDWs’ empowerment and social visibility. Recommendations from the studies suggest that organizations should offer personalized support, including counselling, skills training, and cultural programs, to bolster the resilience, skills, and collective empowerment of these workers. Additionally, the direct impacts of policies governing foreign workers, particularly domestic helpers, on their working conditions and health-related behaviours were underscored. Overall, these studies highlight the importance of addressing psychosocial factors, providing tailored support, and implementing inclusive policies to enhance the health and well-being of FDWs in the Asia Pacific region.

However, despite the growing number of publications, there are still knowledge gaps that require attention. These research gaps emphasize the need for researchers and policymakers to develop informed strategies that target the psychosocial and health-related needs of FDWs more effectively. The study also highlighted the diverse range of countries contributing to research on FDWs, indicating collaborative efforts in understanding and addressing the challenges faced by this population. The analysis of influential studies and sources emphasized the importance of cross-collaboration among countries to share knowledge, expertise, and resources in studying FDWs’ psychosocial and health-related behaviours in the Asia Pacific region. International cross-collaboration among prolific researchers in Asia holds significant importance for expanding knowledge on FDWs’ issues in the region.

Key themes and concepts, such as mental health, COVID-19, and quality of life, were identified as crucial for understanding the experiences of FDWs in the Asia Pacific region. These findings underscore the need for continued research in the field of FDW studies to gain a comprehensive understanding of the complexities involved in the lives of FDWs and to develop effective policies, interventions, and programs that support their well-being.

A limitation of the study is that it relied on utilizing the SCOPUS database and snowballing references, which may have resulted in the exclusion of relevant studies. While these methods can provide a comprehensive overview of the research landscape, they are not exhaustive and may inadvertently overlook certain studies that were not indexed in SCOPUS or identified using the snowballing process. This could potentially lead to a partial representation of the literature and may result in certain knowledge gaps being overlooked or underestimated.

Nonetheless, this study contributes to the current understanding of FDWs’ experiences in the Asia Pacific region and establishes a foundation for further advancements in research and support for this workforce. The insights gained from the bibliometric analyses serve as a valuable resource for researchers, policymakers, and organizations working towards promoting the well-being and safeguarding the rights of FDWs.

## Figures and Tables

**Figure 1 healthcare-12-00690-f001:**
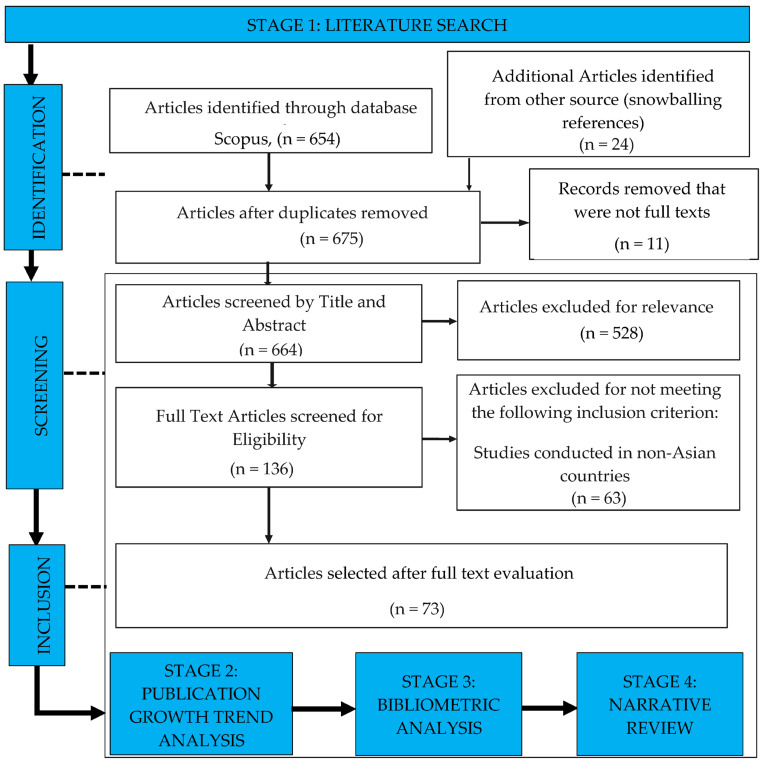
The Workflow of this Study.

**Figure 2 healthcare-12-00690-f002:**
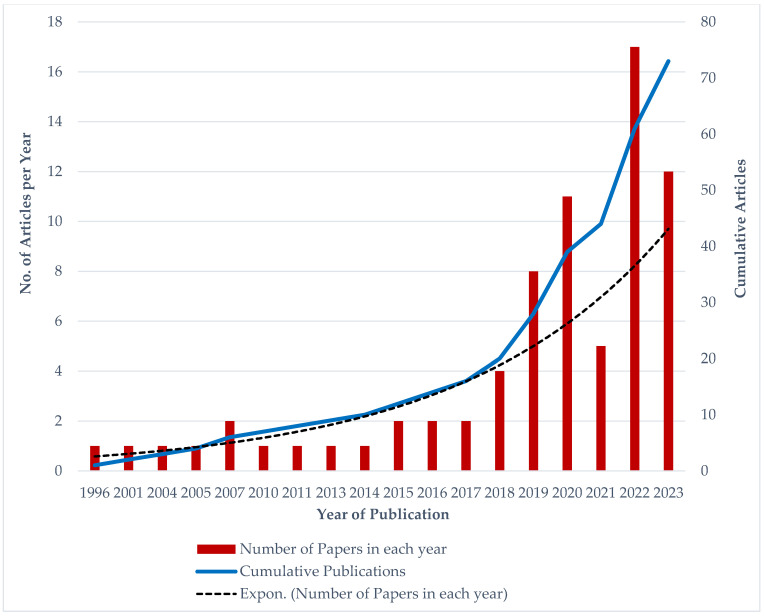
Annual Publication Trend of studies on FDW.

**Figure 3 healthcare-12-00690-f003:**
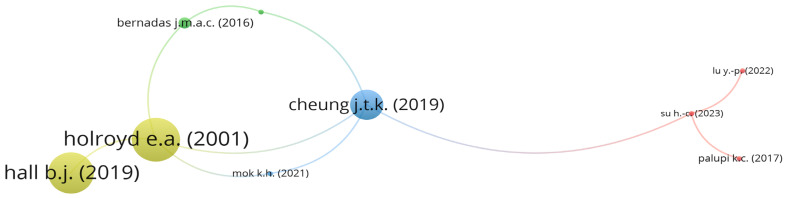
Citation relationship network of studies on FDW [12,17,23,43,48,49,69,74].

**Figure 4 healthcare-12-00690-f004:**
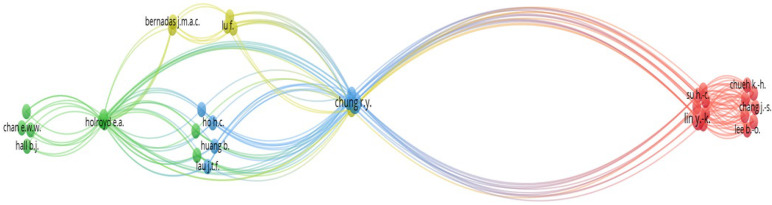
Relationship network of Authors and co-authors who have published studies on FDWs.

**Figure 5 healthcare-12-00690-f005:**
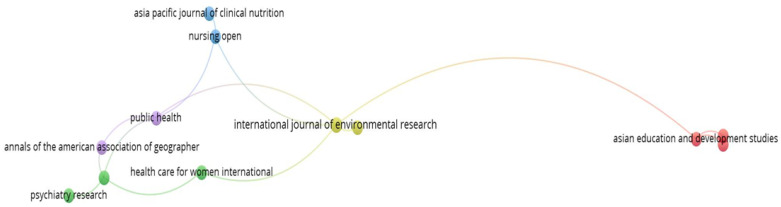
Source and citations relationship of the sources that published articles on FDW.

**Figure 6 healthcare-12-00690-f006:**
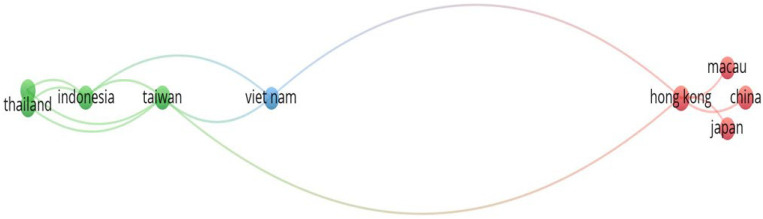
Countries’ collaborative network on FDW studies.

**Figure 7 healthcare-12-00690-f007:**
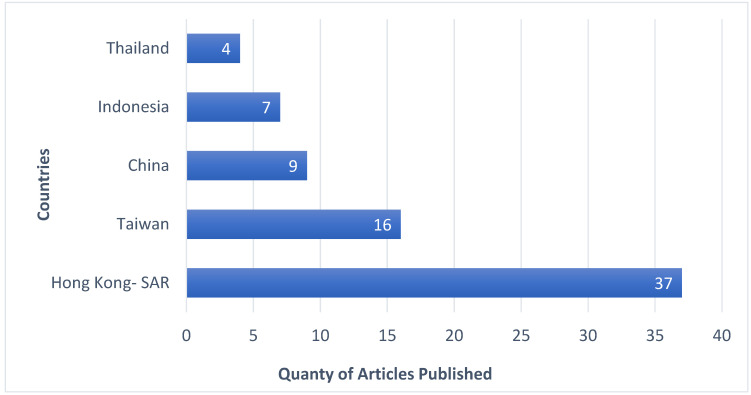
Top Countries that published at least four articles on FDW.

**Figure 8 healthcare-12-00690-f008:**
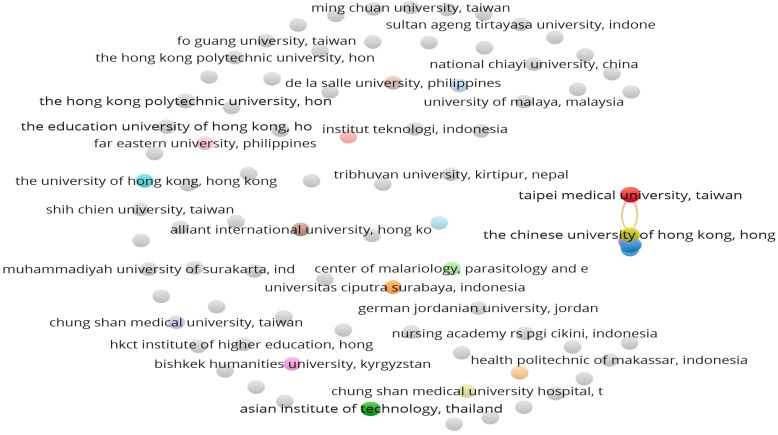
Citation network of the organisations affiliated with the 58 included studies.

**Figure 9 healthcare-12-00690-f009:**
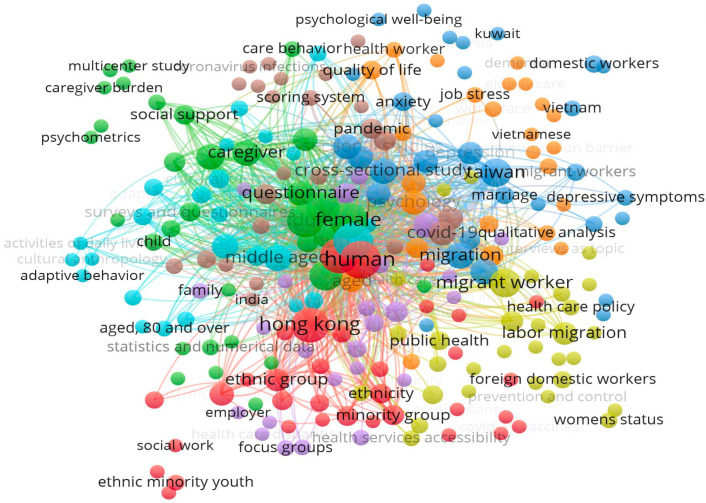
Keywords Co-occurrence Network.

**Figure 10 healthcare-12-00690-f010:**
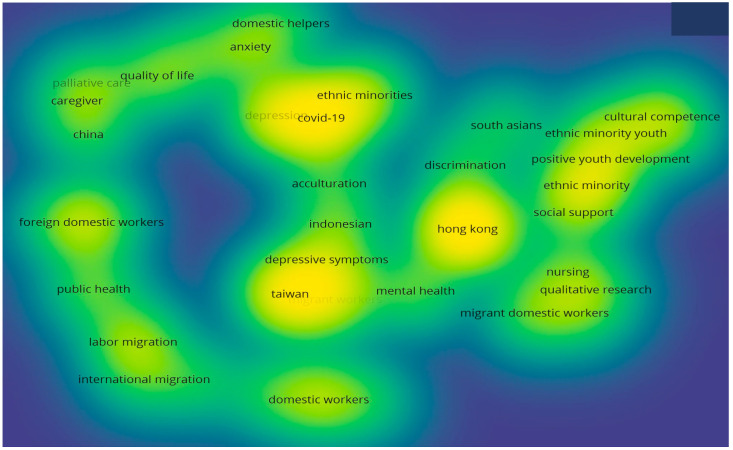
Density visualization map of most active authors’ keywords.

**Table 1 healthcare-12-00690-t001:** List of Included Studies.

Ref.	Authors	Year of Publication	Ref.	Reference	Year of Publication
[21]	Mackenzie A.E., Holroyd E.E.	1996	[22]	Kwok K.	2020
[23]	Holroyd E.A., Molassiotis A., Taylor-Pilliae R.E.	2001	[24]	Vandan N., Wong J.Y.-H., Lee J.J.-J., Yip P.S.-F., Fong D.Y.-T.	2020
[25]	Ogaya C.	2004	[26]	Cheng H.-L., Leung D.Y.P., Ko P.S., Chung M.W., Lam W.M., Lam P.T., Luk A.L., Lam S.C.	2021
[27]	Liang H.-F., Wu K.-M.	2005	[28]	Katigbak J.J.P., Roldan M.D.G.Z.	2021
[29]	Lo H.Y.L.K., Luo X.Y., Lau C.M.J., Wong K.Y.L.	2007	[30]	Deng J.-B., Wahyuni H.I., Yulianto V.I.	2021
[31]	Lee M.-D.	2007	[12]	Mok K.H., Ho H.C.	2021
[32]	Lai M.-Y.	2010	[19]	Gu M.M., Han Y.	2021
[33]	Fan C.C., Chen Y.-Y.	2011	[34]	Kwak Y.K., Wang M.S.	2022
[35]	Ngan L.L.-S., Chan K.-W.	2013	[36]	Lan P.-C.	2022
[37]	Li D., Tsui M.C.M., Tsang H.W.H.	2014	[38]	Wong C.L., Leung A.W.Y., Chan D.N.S., Chow K.M., Chan C.W.H., Ng M.S.N., So W.K.W.	2022
[39]	Phuong N.Q., Venkatesh S.	2015	[40]	Lee W.-C., Chanaka N.S., Tsaur C.-C., Ho J.-J.	2022
[41]	Wui M.G.L., Delias D.	2015	[42]	Oktavianus J., Sun Y., Lu F.	2022
[43]	Bernadas J.M.A.C., Jiang L.C.	2016	[44]	Phuong N.Q., Venkatesh S.	2022
[45]	Altybassarova M., Kozhamzharova M., Beisembayeva G., Amangaliyeva Z., Traissova T.	2016	[46]	Wu C.-Y., Li Y.-Y., Lyver M.J.	2022
[20]	So W.K.W., Wong C.L., Chow K.M., Chen J.M.T., Lam W.W.T., Chan C.W.H., Choi K.C.	2017	[47]	Sumerlin T.S., Kim J.H., Wang Z., Hui A.Y.-K., Chung R.Y.	2022
[48]	Palupi K.C., Shih C.-K., Chang J.-S.	2017	[49]	Lu Y.-P., Lee B.-O., Liu C.-K., Chueh K.-H.	2022
[50]	Arat G., Wong P.W.C.	2018	[51]	Kunpeuk W., Julchoo S., Phaiyarom M., Sinam P., Pudpong N., Loganathan T., Yi H., Suphanchaimat R.	2022
[52]	Shim K., Lee Y.-S.	2018	[53]	Pangaribuan S.M., Lin Y.-K., Lin M.-F., Chang H.-J.	2022
[13]	Wang C.-H., Chung C.-P., Hwang J.-T., Ning C.-Y.	2018	[54]	Chan D.N.S., Choi K.C., Wong C.L., So W.K.W., Fan N.	2022
[55]	Kwok K., Lee K.-M., Law K.-Y.	2018	[56]	Kwan C.K., Lo K.C.	2022
[57]	Arat G., Kerelian N.N.	2019	[58]	Rohmaniyah I., Indiyanto A., Prasojo Z., Julaekhah J.	2022
[59]	Izziyana W.V., Absori, Harun, Wardiono K., Muin F., Dimyati K., Bahtiar	2019	[60]	Tai Y.-S., Yang H.-J.	2022
[61]	Vandan N., Wong J.Y.H., Fong D.Y.T.	2019	[15]	Gu M.M., Chiu M.M., Li Z.	2022
[62]	Pongvongsa T., Nonaka D., Iwagami M., Soundala P., Khattignavong P., Xangsayarath P., Nishimoto F., Kobayashi J., Hongvanthon B., Brey P.T., Kano S.	2019	[63]	Troy C., Tjin A., Perez C J.J., Liu J.	2022
[64]	AlTaher B.B.	2019	[65]	Ho J., Sewell A.	2023
[66]	Ho C.-H., Chan T.-K., Leung N.A.T.C., Leung R., Fan K.-L., Leung L.-P.	2019	[67]	Kwok K.	2023
[17]	Hall B.J., Pangan C.A.C., Chan E.W.W., Huang R.L.	2019	[68]	Lai A.H.-Y., Wang J.Z., Singh A., Wong E.L.-Y., Wang K., Yeoh E.-K.	2023
[69]	Cheung J.T.K., Tsoi V.W.Y., Wong K.H.K., Chung R.Y.	2019	[70]	Yang C.	2023
[71]	Ma C.M.S.	2020	[72]	Asri Y., Chuang K.-Y.	2023
[73]	Yeung N.C.Y., Huang B., Lau C.Y.K., Lau J.T.F.	2020	[74]	Su H.-C., Hu S.H., Chi M.-J., Lin Y.-K., Wang C.-Y., Nguyen T.V., Chuang Y.-H.	2023
[16]	Wong C.L., Chen J., Chow K.M., Law B.M.H., Chan D.N.S., So W.K.W., Leung A.W.Y., Chan C.W.H.	2020	[75]	Lai Y., Fong E.	2023
[76]	Wong C.S.C., Zelman D.C.	2020	[77]	Lui H.-K.	2023
[78]	Lim W., Visaria S.	2020	[79]	Arat G., Wong P.W.-C.	2023
[18]	Leung D.Y.P., Chan H.Y.L., Chiu P.K.C., Lo R.S.K., Lee L.L.Y.	2020	[80]	Sharma A., Adhikari R., Parajuli E., Buda M., Raut J., Gautam E., Adhikari B.	2023
[81]	Suen L.K.P., Rana T.	2020	[82]	Silitonga H.T.H., Winarso H., I’tishom R.	2023
[83]	Chui C.H.-K., Arat G., Chan K., Wong P.W.C.	2020	[84]	Siu J.Y.-M., Cao Y., Shum D.H.K.	2023
[85]	Ju B., Sandel T.L.	2020			

**Table 2 healthcare-12-00690-t002:** Documents with more than 10 citations.

Rank	Article Title	Authors	Year	No. of Citations	Average Citations Per Year	Links	Ref.
1	Filipino domestic workers in Hong Kong: Health-related behaviours, health locus of control and social support	Holroyd E.A., Molassiotis A., Taylor-Pilliae R.E.	2001	51	1.00	4	[23]
2	The effect of discrimination on depression and anxiety symptoms and the buffering role of social capital among female domestic workers in Macao, China	Hall B.J., Pangan C.A.C., Chan E.W.W., Huang R.L.	2019	48	0.44	1	[17]
3	Knowledge, attitudes and practices towards COVID-19 among ethnic minorities in Hong Kong	Wong C.L., Chen J., Chow K.M., Law B.M.H., Chan D.N.S., So W.K.W., Leung A.W.Y., Chan C.W.H.	2020	36	0.28	1	[16]
4	Abuse and depression among Filipino foreign domestic helpers. A cross-sectional survey in Hong Kong	Cheung J.T.K., Tsoi V.W.Y., Wong K.H.K., Chung R.Y.	2019	35	0.32	2	[69]
5	Dancing to different tunes: Performance and activism among migrant domestic workers in Hong Kong	Lai M.	2010	33	1.00	1	[32]
6	Source of social support and caregiving self-efficacy on caregiver burden and patient’s quality of life: A path analysis on patients with palliative care needs and their caregivers	Leung D.Y.P., Chan H.Y.L., Chiu P.K.C., Lo R.S.K., Lee L.L.Y.	2020	27	0.21	0	[18]
7	An exploration of the carers’ perceptions of caregiving and caring responsibilities in Chinese families	Mackenzie A.E., Holroyd E.E.	1996	22	1.00	0	[21]
8	The uptake of cervical cancer screening among South Asians and the general population in Hong Kong: A comparative study	So W.K.W., Wong C.L., Chow K.M., Chen J.M.T., Lam W.W.T., Chan C.W.H., Choi K.C.	2017	19	0.76	1	[20]
9	Exploring family language policy and planning among ethnic minority families in Hong Kong: through a socio-historical and processed lens	Gu M.M., Han Y.	2021	18	0.53	0	[19]
10	Accessing health care: Experiences of South Asian ethnic minority women in Hong Kong	Vandan N., Wong J.Y.H., Fong D.Y.T.	2019	18	0.17	0	[61]
11	Filipino domestic workers and the creation of new subjectivities	Ogaya C.	2004	16	1.00	1	[25]
12	Factors associated with care burden and quality of life among caregivers of the mentally ill in Chinese society	Fan C.C., Chen Y.-Y.	2011	15	1.00	0	[33]
13	Feeling anxious amid the COVID-19 pandemic: Psychosocial correlates of anxiety symptoms among Filipina domestic helpers in Hong Kong	Yeung N.C.Y., Huang B., Lau C.Y.K., Lau J.T.F.	2020	14	0.11	2	[73]
14	An Outsider is Always an Outsider: Migration, Social Policy and Social Exclusion in East Asia	Ngan L.L.-S., Chan K.-W.	2013	14	1.00	0	[35]
15	Caregivers expressed emotion as a mediator of the relationship between neuropsychiatric symptoms of dementia patients and caregiver mental health in Hong Kong	Wong C.S.C., Zelman D.C.	2020	13	0.10	0	[76]
16	“Of and beyond medical consequences”: Exploring health information scanning and seeking behaviours of Filipino domestic service workers in Hong Kong	Bernadas J.M.A.C., Jiang L.C.	2016	13	1.00	1	[43]
17	Growing Old as a Member of an Ethnic Minority in Hong Kong: Implications for an Inclusive Long-Term Care Policy Framework	Chui C.H., Arat G., Chan K., Wong P.W.C.	2020	12	0.09	0	[83]
18	The Foreign Domestic Workers in Singapore, Hong Kong, and Taiwan: Should Minimum Wage Apply to Foreign Domestic Workers?	Wang C.-H., Chung C.-P., Hwang J.-T., Ning C.	2018	11	0.37	0	[13]
19	Risk Preferences and Immigration Attitudes: Evidence from Four East Asian Countries	Shim K., Lee Y.-S.	2018	10	0.33	0	[52]
20	Correlates of consequences of intergenerational caregiving in Taiwan	Lee M.	2007	10	0.91	0	[31]

**Table 3 healthcare-12-00690-t003:** Top Authors with more than one article publication on FDW.

SN	Author	Documents	Citations	Average Citations per Documents	Total Link Strength	Ref
1	So W.K.W.	4	62	15.50	27	[16,20,32,38]
2	Wong C.L.	4	62	15.50	27	[16,20,32,38]
3	Arat G.	4	19	4.75	1	[50,57,79,83]
4	Chan C.W.H.	3	61	20.33	24	[16,20,38]
5	Chow K.M.	3	61	20.33	24	[16,20,38]
6	Chan D.N.S.	3	43	14.33	16	[16,38,54]
7	Kwok K.	3	9	3.00	0	[22,55,67]
8	Choi K.C.	2	20	10.00	20	[20,54]
9	Chung R.Y.	2	37	18.50	1	[47,69]
10	Gu M.M.	2	20	10.00	0	[15,19]
11	Leung A.W.Y.	2	42	21.00	11	[16,38]
12	Leung D.Y.P.	2	30	15.00	0	[18,26]
13	Lin Y.-K.	2	3	1.50	1	[53,74]
14	Phuong N.Q.	2	7	3.50	2	[39,44]
15	Vandan N.	2	26	13.00	11	[24,61]
16	Venkatesh S.	2	7	3.50	2	[39,44]
17	Wong P.W.C.	2	18	9.00	1	[50,83]

**Table 4 healthcare-12-00690-t004:** Top Journals where at least two articles on FDW were published.

SN	Source Title	Documents	Citations	Average Citations per Documents	Total Link Strength	CiteScore Tracker (2023) [91]
1	International Journal of Environmental Research and Public Health	13	93	7.15	2	6.6
2	Asian and Pacific Migration Journal	3	20	6.67	0	1.0
3	Asian Education and Development Studies	2	7	3.50	2	5.1
4	International Journal of Sociology and Social Policy	2	7	3.50	1	5.3
5	Journal of Applied Gerontology	2	14	7.00	1	4.7

**Table 5 healthcare-12-00690-t005:** Bibliometric analysis of Citations per Country.

SN	Country/Territory	Quantity of Articles	Percentage of Total Documents (%)	Number of Citations	Average Citation Per Document	Total Link Strength (TLS)	Nominal GDP Rank ^a^
1	Hong Kong SAR	37	41.11	387	10.46	6	40th
2	Taiwan	16	17.78	60	3.75	7	22nd
3	China	9	10.00	33	3.67	2	2nd
4	Indonesia	7	7.78	16	2.29	5	16th
5	Thailand	4	4.44	12	3.00	3	30th
6	Japan	2	2.22	20	10.00	1	4th
7	Macau	2	2.22	52	26.00	1	102nd
8	Singapore	2	2.22	11	5.50	0	32nd
9	South Korea	2	2.22	10	5.00	0	13th
10	India	1	1.11	1	1.00	3	5th
11	Jordan	1	1.11	1	1.00	0	93rd
12	Kazakhstan	1	1.11	0	0.00	0	52nd
13	Kyrgyzstan	1	1.11	0	0.00	0	148th
14	Laos	1	1.11	4	4.00	0	144th
15	Malaysia	1	1.11	5	5.00	0	36th
16	Nepal	1	1.11	0	0.00	0	100th
17	Philippines	1	1.11	1	1.00	0	34th
18	Vietnam	1	1.11	0	0.00	4	35th

^a^ Nominal GDP Rank was sourced from the International Monetary Fund 2021 estimates, World Economic Outlook Database, October 2021 [92].

**Table 6 healthcare-12-00690-t006:** Top Organizations with more than two documents on FDW.

No.	Organization	Country	Quantity of Documents	Number of Citations	Av. Citation per Document	Total Link Strength
1	The Chinese University of Hong Kong	Hong Kong	12	249	20.75	14
2	The University of Hong Kong	Hong Kong	11	102	9.27	10
3	The Hong Kong Polytechnic University	Hong Kong	6	49	8.17	0
4	The Education University of Hong Kong	Hong Kong	4	24	6.00	0
5	Taipei Medical University	Taiwan	4	9	2.25	2
6	Lingnan University	Hong Kong	4	2	0.50	1
7	City University of Hong Kong	Hong Kong	3	18	6.00	3
8	University of Macau	Macau	2	52	26.00	1
9	Caritas Institute of Higher Education	Hong Kong	2	22	11.00	3
10	Asian Institute of Technology	Thailand	2	7	3.50	1
11	Universitas Gadjah Mada	Indonesia	2	7	3.50	1
12	National Yang Ming Chiao Tung University	Taiwan	2	3	1.50	2

**Table 7 healthcare-12-00690-t007:** Most active keywords with a minimum of three occurrences.

SN	Keyword	Occurrences	Total Link Strength
1	COVID-19	8	10
2	Depression	5	8
3	Ethnic Minority	5	3
4	Domestic Workers	4	3
5	Ethnic Minorities	4	8
6	Foreign Domestic Workers	4	3
7	Labour Migration	4	3
8	Migrant Workers	4	6
9	Anxiety	3	6
10	Cultural Competence	3	2
11	Depressive Symptoms	3	6
12	Ethnic Minority Youth	3	2
13	Mental Health	3	5
14	Migrant Domestic Workers	3	3
15	Public Health	3	2
16	Quality of Life	3	3

**Table 8 healthcare-12-00690-t008:** Study characteristics from the top-cited studies.

Rank	Article Title	Country of Study	Study Type	Study Participants	Ref.
1	Filipino domestic workers in Hong Kong: Health-related behaviours, health locus of control and social support	Hong Kong SAR-China	Cross-sectional survey	290 female Filipino	[23]
2	The effect of discrimination on depression and anxiety symptoms and the buffering role of social capital among female domestic workers in Macao, China	Macao, China	Survey	131 female Filipino	[17]
3	Knowledge, attitudes and practices towards COVID-19 among ethnic minorities in Hong Kong	Hong Kong SAR-China	Cross-sectional survey	352 South Asians	[16]
4	Abuse and depression among Filipino foreign domestic helpers. A cross-sectional survey in Hong Kong	Hong Kong SAR-China	Cross-sectional survey	105 Filipino	[69]
5	Dancing to different tunes: Performance and activism among migrant domestic workers in Hong Kong	Hong Kong SAR-China	Interviews	267,778 Indonesians and Filipinas	[32]
6	Source of social support and caregiving self-efficacy on caregiver burden and patient’s quality of life: A path analysis on patients with palliative care needs and their caregivers	Hong Kong SAR-China	Cross-sectional survey	225 South Asians	[18]
7	An exploration of the carers’ perceptions of caregiving and caring responsibilities in Chinese families	Hong Kong SAR-China	Semi-structured interviews	10 South Asian women	[21]
8	The uptake of cervical cancer screening among South Asians and the general population in Hong Kong: A comparative study	Hong Kong SAR-China	Cross-sectional survey	161 South Asian women	[20]
9	Exploring family language policy and planning among ethnic minority families in Hong Kong: through a socio-historical and processed lens	Hong Kong SAR-China	Interviews	10 South Asian women	[19]
10	Accessing health care: Experiences of South Asian ethnic minority women in Hong Kong	Hong Kong SAR-China	Interviews	30 South Asian women	[61]
11	Filipino domestic workers and the creation of new subjectivities	Hong Kong SAR-China and Singapore	Interviews	40 female Filipino	[25]
12	Factors associated with care burden and quality of life among caregivers of the mentally ill in Chinese society	Taiwan, China	Cross-sectional survey	90 South Asians	[33]
13	Feeling anxious amid the COVID-19 pandemic: Psychosocial correlates of anxiety symptoms among Filipina domestic helpers in Hong Kong	Hong Kong SAR-China	Cross-sectional survey	295 female Filipino	[73]
14	An Outsider is Always an Outsider: Migration, Social Policy and Social Exclusion in East Asia	South Korea, Taiwan, Hong Kong SAR-China, mainland China,	Interviews	56 Southeast and East Asian	[35]
15	Caregivers expressed emotion as a mediator of the relationship between neuropsychiatric symptoms of dementia patients and caregiver mental health in Hong Kong	Hong Kong SAR-China	Cross-sectional survey	89 South Asian	[76]
16	“Of and beyond medical consequences”: Exploring health information scanning and seeking behaviours of Filipino domestic service workers in Hong Kong	Hong Kong SAR-China	Interviews	23 female Filipino	[43]
17	Growing Old as a Member of an Ethnic Minority in Hong Kong: Implications for an Inclusive Long-Term Care Policy Framework	Hong Kong SAR-China	Semi-structured interviews	30 Nepalese	[83]
18	The Foreign Domestic Workers in Singapore, Hong Kong, and Taiwan: Should Minimum Wage Apply to Foreign Domestic Workers?	Singapore, Hong Kong, and Taiwan	Secondary data	2 million Southeast and East Asian	[13]
19	Risk Preferences and Immigration Attitudes: Evidence from Four East Asian Countries	China, Japan, Korea, and Taiwan	Cross-sectional survey	8417 Southeast and East Asian	[52]
20	Correlates of consequences of intergenerational caregiving in Taiwan	Taiwan, China	Cross-sectional survey	130 South Asian	[31]

## Data Availability

The data that support the findings of this study are available from the corresponding author upon reasonable request.

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
