# Peer review of "Research Trends of Studies on Psychosocial and Health-Related Behaviours of Foreign Domestic Workers in Asia Pacific: A Bibliometric Analysis"

_healthcare, 2024, doi:10.3390/healthcare12060690_

Round 1

Reviewer 1 Report

Comments and Suggestions for Authors

This study utilized bibliometric analyses to examine research trends on the psychosocial and health-related behaviours of FDWs in the Asia Paciffc region.  The analysis identiffed inffuential studies, active authors, sources with high publication numbers, the geographical distribution of studies, the countries and organizations, key themes and concepts, and the most active keywords,  which can provide other researchers with an overview of the field. 

However, the paper does not make a meta-analysis or review of the substance of the existing researches, such as status quo, causes, trends, and coping strategies in the psychosocial and health-related behaviours of FDWs in the Asia Paciffc region. It is recommended to supplement these contents, just like the analysis of keywords.

Author Response

Dear Reviewer,

Comment: This study utilized bibliometric analyses to examine research trends on the psychosocial and health-related behaviours of FDWs in the Asia Paciffc region.  The analysis identiffed inffuential studies, active authors, sources with high publication numbers, the geographical distribution of studies, the countries and organizations, key themes and concepts, and the most active keywords,  which can provide other researchers with an overview of the field. 

However, the paper does not make a meta-analysis or review of the substance of the existing researches, such as status quo, causes, trends, and coping strategies in the psychosocial and health-related behaviours of FDWs in the Asia Paciffc region. It is recommended to supplement these contents, just like the analysis of keywords.

Response: A narrative review of the key findings from the studies has been added to the study, specifically sections on methodology, results and discussion.

Reviewer 2 Report

Comments and Suggestions for Authors

I want to express my gratitude for giving me the opportunity to review this manuscript. The time spent creating and submitting it is greatly appreciated. However, I believe that there are several areas that require improvement for manuscript’s potential publication.   

Introduction:

1)     The introduction lacks research on the topic under study. Please expand the literature.

2)     The relevance of the study, before presenting the hypothesis, should be better articulated et the end of the Introduction section.

Materials and Methods:

1)     Statistical analysis: Why didn't the authors use a review or systematic review, or a meta-analysis?

Results:

1)     Why were only studies conducted in Asia included?"

Discussion:

1)     Please integrate the study results with other research findings present in the literature.

2)     What are the practical implications?

Author Response

Dear Reviewer,

Introduction:

1)     The introduction lacks research on the topic under study. Please expand the literature.

Response: The introduction has been expanded to include the research.

2)     The relevance of the study, before presenting the hypothesis, should be better articulated et the end of the Introduction section.

Response: The last paragraph has been refined.

Materials and Methods:

1)     Statistical analysis: Why didn't the authors use a review or systematic review, or a meta-analysis?

Response: These information has been added. “Bibliometric analysis was employed in this study to provide a statistical evaluation and visualization of the research trend in the field of studies on FDWs in the Asia Pacific region, thereby offering a unique perspective that complements traditional review methods such as systematic review or meta-analysis.”

Results:

1)     Why were only studies conducted in Asia included?"

Response: This information has been added. “Only studies conducted in Asia were included in the research selection process to ensure that the analysis focused on the context of psychosocial and health-related behaviors of FDWs in the Asia Pacific region. By narrowing the scope to Asian countries, the study aimed to capture the unique context and challenges faced by FDWs in this specific geo-graphical area. This approach thus allowed for a more targeted exploration of the research literature that directly pertains to the experiences and needs of FDWs in Asia.”

Discussion:

1)     Please integrate the study results with other research findings present in the literature.

Response: The discussion section now incorporated other research findings.

2)     What are the practical implications?

Response: This information has been added. “The generated bibliometric maps and analyses offer valuable guidance for researchers and academics interested in the field of FDWs. By identifying the most active keywords, impactful documents, authors, and organizations, this information can guide researchers on the current trends, key areas of focus, and potential collaborators within the field. Understanding the citation relationships and impact of studies can help direct future research endeavours and collaborations. Additionally, identifying the most discussed topics and trends related to FDWs' psychosocial and health-related behaviours can aid in formulating evidence-based policies, interventions, and programs aimed at improving the well-being and rights of FDWs in the Asia Pacific region. Thus, researchers, practitioners, and policymakers can leverage these highly cited articles and impactful studies to build upon existing knowledge, promote collaboration, and advocate for the rights and well-being of FDWs.”

Reviewer 3 Report

Comments and Suggestions for Authors

Thank you for the opportunity to review this manuscript. In order for it to reach the standard required of the Journal I suggest a few edits as follows:

1. Table 5: Ensure the words in the titles are all on one line perhaps reduce the font size to ensure they fit. 

2. Figure 1: Stage 2 refers to a Trend Analysis. This analysis should also be referred to in the abstract and the authors need to state clearly how this analysis was conducted. 

2. Discussion: The discussion commences with the themes? However, it is unclear how these themes were generated and appear to be only from a few of the articles. The method for conducting a thematic analysis needs to be clearly stated within the methods section.  The discussion section is the first indication that there was a thematic analysis of the 73 articles??

There needs to be a greater critical analysis of the articles and the relationships between the various authors with greater attention paid to the key themes and concepts.  There is no analysis of the methods used within these 73 studies to measure psychosocial and health related behaviours and indeed at times mental health is used interchangeably with psychosocial. These two concepts are completely different. 

Author Response

Dear Reviewer,

  1. Table 5:Ensure the words in the titles are all on one line perhaps reduce the font size to ensure they fit. 

Response: This has been corrected.

  1. Figure 1:Stage 2 refers to a Trend Analysis. This analysis should also be referred to in the abstract and the authors need to state clearly how this analysis was conducted. 

Response: Trend analysis has been corrected to Publication growth trend analysis. This has been clearly stated also in the abstract.

  1. Discussion:The discussion commences with the themes? However, it is unclear how these themes were generated and appear to be only from a few of the articles. The method for conducting a thematic analysis needs to be clearly stated within the methods section.  The discussion section is the first indication that there was a thematic analysis of the 73 articles??

Response: this information has been added “The analysis of top-cited studies involved a narrative review that delved into the research themes, the countries where the studies were conducted, the study designs employed, and the key findings.”

There needs to be a greater critical analysis of the articles and the relationships between the various authors with greater attention paid to the key themes and concepts.  There is no analysis of the methods used within these 73 studies to measure psychosocial and health related behaviours.

Response: Analysis of the studies exploring the key themes on psychosocial and health related behaviours has been added.

Indeed at times mental health is used interchangeably with psychosocial. These two concepts are completely different. 

Response: This has been corrected.

We are grateful for all the constructive comments which have contributed in enhancing the quality of this manuscript. 

Reviewer 4 Report

Comments and Suggestions for Authors

The paper contains bibliometric analyses examining research trends on the psychosocial and health-related behaviours of foreign domestic workers in the Asia Pacific region. The analyses are properly conducted, but originality and of the paper should assessed as low, and its importance for the development of knowledge concerning foreign domestic workers in Asia is marginal. The paper shows only growing interest in the subject (which is undoubtly important in Asia Pacific region) from social point of view) in recent years. The presentation of hitherto reasearch themes (pp. 19–20) and potential gaps in literaturę (pp. 20–21) could be somewhat useful for further research.

Author Response

Dear Reviewer,

Comment: The paper contains bibliometric analyses examining research trends on the psychosocial and health-related behaviours of foreign domestic workers in the Asia Pacific region. The analyses are properly conducted, but originality and of the paper should assessed as low, and its importance for the development of knowledge concerning foreign domestic workers in Asia is marginal. The paper shows only growing interest in the subject (which is undoubtly important in Asia Pacific region) from social point of view) in recent years. The presentation of hitherto reasearch themes (pp. 19–20) and potential gaps in literaturę (pp. 20–21) could be somewhat useful for further research.

Response: Thank you for your feedback. Further revision has been made to enhance the quality of this study.

Round 2

Reviewer 1 Report

Comments and Suggestions for Authors

The data analysis method of the original manuscript is novel and effective. According to my suggestions, the authors have added to the revised version a discussion of the substance of the existing researches and enhanced the value of the paper. Therefore, this paper will be a good reference for researchers in this field, and I think it has met the publication requirements of the journal.

I recommend that authors add their own comments on the substance of existing studies and conduct a replication review to avoid duplication of existing studies as much as possible.

Author Response

Dear Reviewer,

Comments: 

The data analysis method of the original manuscript is novel and effective. According to my suggestions, the authors have added to the revised version a discussion of the substance of the existing researches and enhanced the value of the paper. Therefore, this paper will be a good reference for researchers in this field, and I think it has met the publication requirements of the journal.

I recommend that authors add their own comments on the substance of existing studies and conduct a replication review to avoid duplication of existing studies as much as possible.

Response:

The substance of existing studies has been added

“The included studies shed light on pivotal insights concerning the psychosocial and health-related behaviors of FDWs in the Asia Pacific region. Notably, psychosocial factors such as engaging in social activities beyond their employers' households were identified as crucial determinants of the well-being of FDWs, emphasizing their significance on FDWs’ empowerment and social visibility. Recommendations from the studies suggest that organizations should offer personalized support, including counseling, skills training, and cultural programs, to bolster the resilience, skills, and collective empowerment of these workers. Additionally, the direct impacts of policies governing foreign workers, particularly domestic helpers, on their working conditions and health-related behaviors were underscored. Overall, these studies highlight the importance of addressing psychosocial factors, providing tailored support, and implementing inclusive policies to enhance the health and well-being of FDWs in the Asia Pacific region."

A report on replication review has been added

“The literature reveals a substantial body of research focused on the psychosocial and health-related behaviors of foreign domestic workers (FDWs) in the Asia Pacific region. Previous studies have undertaken scoping reviews, systematic reviews, and me-ta-analyses examining various aspects of FDWs' well-being. For instance, a scoping re-view delved into the health stressors, problems, and coping mechanisms of migrant domestic workers worldwide [60]. Additionally, Ho and colleagues [61] explored peer support and mental health among migrant domestic workers globally through a scoping review. Perski and colleagues [62] conducted a systematic review and meta-analysis of ecological momentary assessment studies on five key health behaviors on a global scale, albeit not specifically focusing on FDWs. Notably, there was a lack of bibliometric reviews in the existing literature. This study aims to address this gap by conducting a comprehensive bibliometric analysis on psychosocial and health-related behaviours of FDWs in Asia Pacific.”

Thank you very much for your help in improving this manuscript further.

Reviewer 2 Report

Comments and Suggestions for Authors

accept in this form

Author Response

Thank you for your help in improving the manuscript.

Reviewer 3 Report

Comments and Suggestions for Authors

Thank you for the corrections. The manuscript reads and flows a lot better. The findings are integrated and now relate to the discussion and conclusions. Well done.

Author Response

Thank you for your help in enhancing the quality of this paper.